# Exploring Health Care Professionals’ Perceptions Regarding Shared Clinical Decision-Making in Both Acute and Palliative Cancer Care

**DOI:** 10.3390/ijerph192316134

**Published:** 2022-12-02

**Authors:** Helena Ullgren, Lena Sharp, Per Fransson, Karin Bergkvist

**Affiliations:** 1Department of Nursing, Umeå University, 901 87 Umeå, Sweden; 2Department of Oncology and Pathology, Karolinska Institute, 171 77 Stockholm, Sweden; 3ME Head & Neck, Lung & Skin Cancer, Karolinska Comprehensive Cancer Center, 171 76 Stockholm, Sweden; 4Regional Cancer Center, 104 25 Stockholm, Sweden; 5Department of Nursing Science, Sophiahemmet University, 114 86 Stockholm, Sweden; 6Department of Neurobiology, Care Sciences and Society, Division of Nursing, Karolinska Institute, 171 77 Stockholm, Sweden

**Keywords:** cancer care, palliative care, clinical decision-making, clinical research home care, focus groups

## Abstract

Developments in cancer care have resulted in improved survival and quality of life. Integration of acute and palliative cancer care is desirable, but not always achieved. Fragmented care is associated with sub-optimal communication and collaboration, resulting in unnecessary care transitions. The aim of this study was to explore how health care professionals, from both acute and palliative care, perceive clinical decision-making when caring for patients undergoing active cancer treatment in parallel with specialized palliative care at home. Methods: Qualitative explorative design, using online focus-group interviews, based on patient-cases, among health care professionals (physicians and nurses) and Framework Analysis. Results: Six online focus-group interviews were performed. Few signs of systematic integration were found, risking fragmented care, and putting the patients in a vulnerable situation. Different aspects of uncertainty related to mandates and goals-of-care impacted clinical decision-making. Organizational factors appeared to hinder mutual clinical decision-making as well as the uncertainty related to responsibilities. These uncertainties seemed to be a barrier to timely end-of-life conversations and clinical decisions on optimal care, for example, the appropriateness of transfer to acute care. Conclusions: Lack of integration between acute and palliative care have negative consequences for patients (fragmented care), health care professionals (ethical stress), and the health care system (inadequate use of resources).

## 1. Introduction

New treatment options and the addition of increased support earlier in the cancer care trajectories, have resulted in more favorable outcomes, such as improved survival [1] and quality of life (QoL) [2]. A consequence is often longer, more intense treatment periods, resulting in extensive treatment and intensity of care at End-of-Life (EOL). This may entail increased levels of health care utilization [3,4]. The developments in cancer treatments may also challenge the complex process of clinical decision-making, e.g., knowing when to stop treatment or change intensity of care [5] and/or predicting the course of disease and prognosis, where contextual factors, patients’ preferences as well as medical conditions need to be interpreted and evaluated to select actions [6].

Early palliative care (introduced within 8 weeks of diagnose, integrated with acute cancer care) have shown improved QoL [2], both fewer unplanned hospital admissions [7,8] and visits to the emergency department (ER) [9]. In a systematic review [10], various ways of providing palliative care were described. However, several of the studies were lacking thorough descriptions of palliative care models [10]. Often palliative care is organized as General palliative care (described as palliative care provided by general practitioners, district nurses or from hospital units), or Specialized palliative care (SPC), provided by a team of health care professionals with training in palliative care, often including inpatient care or SPC at home [11]. Both general and SPC can be provided early in the disease trajectory, as well as at EOL. In Sweden, the provision of palliative care varies between regions [12], both in terms of accessibility and organization. Recent recommendations both in research and by expert panels, focus on the systematic integration of acute and palliative cancer care [13]. Kaasa et al. (2018) suggests several levels of integration, from less formal working in parallel to being fully integrated. The goal is to provide health care organized from the individual patient’s needs, with a seamless provision, formed as one unit from the existing system [13], with clear and integrated goals-of-care.

Discussions on goals at EOL, taking into account the patients preferences, is one of several quality indicators for EOL care [12]. Kuusistio et al., 2020, showed that these EOL discussions however, often were conducted too late [14]. In a recent review, the authors concluded that most patients with cancer preferred, if possible, to die at home [15] and wanted to prepare and plan for EOL [16].

In the context of this study, cancer treatment and follow-up are provided by three acute hospitals in the region of Stockholm. The acute cancer care teams offer support and symptom management, with the main point of contact, usually a specialist nurse, working at the hospital out-patient clinics [17]. During the last decade, access to SPC at home has increased in this region, regardless of disease stage [18]. Some of the SPC units provide home care while others have access to SPC in-patient care. Thus, the acute cancer care and SPC at home care teams are organized in separate organizations [18]. Non-medical support at home is provided, if needed, by the municipalities, again separately organized.

Results from a previous Swedish survey from the same region, [19] indicate sub-optimal communication between the acute and palliative cancer care teams, consequently increasing the risk for unplanned care (ER visits and readmissions) [20], unnecessary care transitions, and unclear professional responsibilities between the teams involved [21].

The objectives were to explore factors that would potentially enable or hinder clinical decision-making when considering for example to send patients in palliative care to ER or not and the dilemmas these situations may involve. Therefore, this study aimed to explore how health care professionals in both acute cancer and palliative care settings, perceive clinical decision-making, regarding patients undergoing both active cancer treatment and concomitant palliative care.

## 2. Materials and Methods

### 2.1. Study Design

An explorative research design was used, including focus group interviews (FGI’s) with health care professionals (nurses and physicians).

### 2.2. Data Collection

As clinical-decision making in acute and palliative cancer care are a highly complex topic, several viewpoints as well as a broad perspective needed to be covered, FGI’s were performed to collect data [22].

Further to enable health care professionals to participate, during the COVID-19 pandemic, as well as get participation from different geographical areas, the FGI’s were conducted digitally [23]. A strategic sample of participants from different teams was approached (i.e., SPC at home teams from different geographical areas and organizations in the region of Stockholm as well as from the acute care hospitals treating cancer), with the purpose of reaching different perspectives.

### 2.3. Interview Guide

An interview guide with fictive patient cases was used to facilitate and structure the FGI´s [24]. Two fictive cases were created (Appendix A), based on both clinical experiences and preliminary findings from previous studies [25,26,27]. These cases, as well as the prompting questions, were developed by the first author (HU) and tested for face validity before data collection by health care professionals (three physicians and three nurses) from both acute and palliative cancer care. After accepting to participate via e-mail, a test interview was performed, to test both the online connection/recording as well as the cases/questions. The cases were slightly altered after the test, according to suggestions. During the FGI’s, prompting questions related to the cases were raised (Appendix A). Purposely the cases did not contain all clinical information, aiming for the participants to suggest and discuss possible enablers and barriers related to clinical decision-making.

### 2.4. Recruitment

An information letter was sent to the Heads of Departments asking for approval. Once approved, the same letter was sent to the unit managers, respectively. The unit managers contacted staff members and informed the researcher when a focus-group participant had approved participation. A calendar invite including a Microsoft Teams^®^ link was sent, with more information regarding the FGI, as well as an informed consent form.

After introductions, the FGI’s started with the participants briefly reading through fictive cases (also shown on the computer screen during the FGI for reference). For all FGIs, 90 min were allocated.

The FGI was conducted by HU (moderator, leading the FGI) and KB (assessor, actively listening, posing follow-up questions, and summarizing the FGI), between 28 January and 21 June 2021. During the FGI´s, field notes were taken by both the moderator (HU) and the assessor (KB). The notes were compared and discussed after each interview, with the focus to reflect how the FGI were conducted, how the participants interacted with each other and what topics were highlighted.

All interviews were recorded and transcribed by an experienced medical secretary, not participating in the FGI´s or being involved in any other parts of the study.

Any names of participants or organizations were changed in the transcription and only anonymized data was stored.

The Consolidated criteria for reporting qualitative research (COREQ) [28], were followed throughout the process. The Regional ethics board approved the study (DNR 2020-00910).

### 2.5. Data Analysis

The Framework Analysis (FA) was chosen to analyze the FGI’s for a flexible and highly transparent approach, offering systematic organization of the data [29,30,31]. Data was organized in a matrix (using Microsoft Excel^®^), enabling collaboration during the analysis between the members of the research team [31]. FA was also chosen to support transparency of the researchers pre-understanding of the studied topic [31,32], aiming to increase trustworthiness as well as rigor. The fact that the analytic process was checked by the two authors not present during the FGI’s (PF and LS) also contributed to the rigor of this study. We followed the FA process with each step interlinked during the analysis (Figure 1). A preliminary framework was identified, applied early and later altered during the analytical process (Appendix B). The analytical process is described below in Figure 1. The field notes were read and reflected upon alongside of the transcripts during the familiarization phase. Data was collected until sufficient information power was reached on the topic explored in this study [33]. The transcripts were checked by the first author, also present at all interviews, but not sent back to participants for checking.

## 3. Results

Six FGI´s (four including SPC teams and two with acute cancer teams) were conducted, with a total of 22 participants, Table 1. The duration of the interviews varied between 72 min to 84 min, (median of 78 min) recordings started after introductions and taking notes on background info. Out of three invited acute cancer care teams, two accepted while four of six invited SPC at home teams participated. Both the acute care and SPC providers that declined, stated lack of time as the reason for not participating. The number of participants varied between two to five between the FGI’s. In four of the six groups both nurses and physicians participated, while one group consisted of only nurses and another of only physicians.

The findings are presented through one main overarching theme, *Uncertainties in decision-making.* Different aspects of uncertainty related to clinical decision-making, in four main themes with associated sub-themes are presented under each heading and in Figure 2. Quotes are marked with “…”. Whenever a quote is shortened a, it is marked with (…). In addition, the quotes are labelled by the number of FGI and as physicians (MD) or nurses (RN).

### 3.1. Uncertainties in Clinical Decision-Making

Throughout the analysis, different features of uncertainty were evident in most aspects of clinical decision-making. Already during the first phase of analysis, when reading transcripts alongside with the field notes, we reflected over uncertainties as an aspect that seemed to impact clinical decision-making. At the final stages of the analysis, the overarching theme became clear, with uncertainties described in all main themes. This overarching theme is linked with all main and sub-themes. This was confirmed during the analytic process by the two authors not present during the FGI’s.

Across the FGIs, uncertainties were reported regarding organizational factors; an inconsistency between organizations (access to care, resources, and competencies), were described to largely impact clinical decisions, i.e., how to act, plan for care activities and interact with patients and their families. In addition, uncertainty of who should make and/or initiate important decisions, are portrayed in a theme, *Organizational factors impacting clinical decision-making*. Other examples of uncertainty related to the patient’s prognostic understanding or the patients living situation also seemed to influence clinical decisions and were reported in, *Patients and informal caregivers’ prerequisites*. In addition, aspects of uncertainty were apparent in a theme related to goals-of-care as well as an uncertainty of the patients’ prognosis, *Balancing patient’s medical condition and needs based on the uncertainty of the future.* These aspects of uncertainty appeared to hinder and impact the clinical decisions, leading to ethical conflicts, presented in the theme, *Balancing ethical dilemmas.*

#### Organizational Factors Impacting Clinical Decision-Making

Organizational factors seemed to impact clinical decisions, several related to different aspects of resources, covered in the sub-theme, *Resources, geography and timing*. Participants from both SPC and acute cancer care described how their resources affected their decisions, e.g., the ratio of nurses/physicians per patient, or whether a physician was on call during weekends or not. As an example, geography seemed to impact clinical decision-making; if a patient lived remotely on an island difficult to access, they were more likely to send the patient to the ER, or at least to the hospital, since frequent visits were hindered by location and travel time. It was also discussed by several SPC teams how the workload (number of patients in need of multiple home visits) impacted decisions on whether to send a patient to ER or not (rather than the medical and/or psychosocial condition).

“*It’s Friday evening and then you cannot expect an assessment by a physician until Monday really, the threshold for sending the patient in is lower, since this is the only way of getting an appropriate assessment.*”(FG2, MD2)

The health care professionals (HCPs) discussed healthcare inequalities and the heterogeneity of resources in the sub-theme, *Uncertainty of the organization*. For example, HCPs in one of the acute care hospitals were reported to have access to a specialized cancer ER, meaning that the patients did not have to go to the general ER, where often other acute conditions are prioritized. In addition, heterogeneity in access to SPC inpatient beds also appeared to impact these decisions. In one sub-theme, the reasoning centered on the heterogeneity of competence within/between SPC teams (creating uncertainty about what to expect); *Access to competent care by competent staff.*

“*I think we all know that the care is very different between SPC teams. Some does all (assessments, antibiotics) and care for them until they are dying, while others always send in the patients.*”(FG5, MD2)

Another finding related to this sub-theme was the resources from either informal caregivers and/or professional care (personal hygiene, feeding), this type of support being separately organized by the municipalities. The SPC teams discussed the difficulties of organizing this important support from the municipalities, especially if urgent, as the process was time-consuming. The HCPs described the challenges involved in coordinating care activities between SPC at home and non-medical support at home.

The negative consequences of sending palliative patients to the acute hospital were highlighted by the acute care teams. One organizational factor described was the routine that if a patient were admitted to the acute care hospital for more than 72 h, a new referral to SPC was needed. The new referral, hindering continuity of care and risking unnecessary acute care, is described in, *System and policy*.

“*..This happens all the time- Friday night, because you are perhaps new- it feels stressful when someone cannot breathe properly. You know it may be a pulmonary embolus and there will be a CT (computer tomography). The patient will be admitted and stay over the weekend. Then they lost their place at SPC at home. And then it will be hard to discharge the patient to home.*”(FG5, MD1)

The health care professionals from both settings, described how they felt restrained in clinical decisions due to how the health care system and how economical incitements impacted. This was expressed several times with frustration, i.e., the pressure of fulfilling the policy to occupy all beds in the palliative inpatient ward (leaving no room for an urgent admission from home and/or hindering continuity of care) and fragmented care consequently.

“*It is frustrating to keep the budget, our ward needs to be fully occupied all the time and when that happens, the patient´s supposed to “choose” another SPC provider. It does matter, it matters a lot. In this case, the patient might end up at a palliative ward somewhere else, making it more difficult for the patients’ family to visit.*”(FG3, RN3)

Among participants from both settings and professions, it seemed to be unclear who had the overarching responsibilities for a patients’ care. In the sub-theme, *Uncertainty of mandate and goals-of-care*, this was discussed in various ways, for instance when a patient recently had completed cancer treatment and had a follow-up acute cancer care appointment scheduled but also ongoing SPC at home.

“*She is (the patient) still in the care of the oncology clinic even if she doesn´t have active treatment, she still has an appointment there. Here we need to be clear, a clearer decision, SPC is responsible, but the oncology clinic is responsible in one way.*”(FG3, RN3)

All participants agreed on shared responsibility in principle, but different views and uncertainty on what this exactly meant were apparent. Responsibility for cancer treatment and symptom management seems to be the theoretical threshold, but in practice, this seems confusing as treatment and symptom management need to be interlinked in person-centered care. The perceived silos of acute versus palliative care appeared to impact a timely EOL conversation (one team waiting for the other to initiate EOL conversations and planning).

“*They, the oncology clinic should have the difficult conversation. We should not do it for them.*”(FG4, RN2)

The apparent lack of clear individual goals of care and thereby the purpose of treatment seems to add to the uncertainty. Another barrier to initiating an EOL conversation was described to be associated with the availability of new treatments and if there was a temporary pause in treatment. This was described by the participants, especially by the SPC teams.

“*during recent years, (...) one doesn’t dare to say no. We are rather thinking, we have a new treatment that possibly could help.*”(FG2, MD1)

The lack of systematic collaboration between the teams was described as a hinder negatively impacting much needed clinical decisions. Participants from both settings expressed that the other care team needed to improve their clinical decision-making and communicate these decisions better.

“*Well, now there´s treatment much longer and tougher. Into the last days. I feel the decision is never made, but perhaps close. The difficult conversation doesn’t happen, it is postponed. And then the patient deteriorates and end up like this (admitted to an acute care hospital at EOL).*”(FG4, RN2)

The SPC participants indicated that a consequence of the poor collaboration between the teams were inadequate clinical decision-making with fragmented care and unnecessary acute admissions. Similar consequences were described by the acute care teams.

“*We are spending time ordering scans and tests, but for this kind of patients it is not just a hospitalization, it is a long journey, hours on a stretcher having bone metastasis and pain. It is so much more. We need to have an adequate plan here.*”(FG5, MD1)

### 3.2. Patients and Informal Caregivers’ Prerequisites

In most of the FGI’s, patients and informal caregiver’s perspective were central in the discussion, both regarding physical conditions, but also their wishes and prognostic understanding. The sub-theme *Patients living situation*, include reasoning related to decision-making in situations if/when the patient was being sent from palliative to acute care. An example described were when a patient suffered from refractory dyspnea and relying heavily on support from informal caregivers. Here, the risk of health inequalities was raised since it was not considered possible to always meet the patient’s wishes to stay at home.

“*(...), in this case you do know she´s living alone. There is clear inequality for people living alone – they don’t receive support the same way.*”(FG1, MD1)

Especially the participants in the SPC teams reasoned regarding patient’s ´wishes and the importance of respecting both the patients and the informal caregiver’s preferences. This was discussed across the FGIs with person-centered approaches as solutions to meet the individual needs of patients and informal caregivers. A sub-theme was identified, related to this topic, *Patient as the decision maker*. Even if both nurses and physicians in acute cancer care teams also discussed examples of patient-centeredness, it seemed more explicit in the SPC FGI’s. It appeared that sometimes the HCPs, perhaps due to feeling out of other strategies, resigned to the patient’s wishes, even in situations that they believed they could manage at home.

“*(…) even if we feel we could handle this at home (for symptom management). We can care for you at home, then this is what will happen, if the patients wish to go to hospital, we cannot say anything else.*”(FG1, RN1)

The latter sub-theme seemed to be linked with another sub-theme, *Patients and Informal caregiver’s health beliefs and prognostic understanding* (such as perceived health and expectations of future and/or cancer treatment). Both SPC and acute cancer care teams agreed that this had an impact on clinical decision-making. Physicians and nurses spoke of how challenging and complex it was when there were discrepancies in prognostic understanding and beliefs between the patient and informal caregivers. The addition of uncertainty of the patients’ prognostic understanding and health beliefs were further complicated by the un-clarity of responsibilities. Even more so when the patient and informal caregivers did not share the same prognostic understanding.

“*(…) you can ask the question why the husband wants this. Is it because of the situation at home, that he feels it is too scary for the kids or is it that he wants her to go to the hospital to be cured.*”(FG3, RN2)

It seemed important, especially in SPC that the patients were made aware of the consequences of remaining at home in these situations. Patients were even expected to take full responsibility for the decision.

“*Many patients don´t want to go to the hospital, they´ve done it before, they know they are not the top priority at ER, they must wait. But the patient must also be aware of what could happen if not going to the hospital. The patient must be prepared to take the consequences!*” (FG4, RN1)

Interestingly, the acute care teams reasoned with a different outlook on the negative consequences of these patients returning to the hospital and the risk of not being able to fulfill the patient’s preferences, in this case, related to the place of death.

“*.. If the patient’s desire is to die at home, the possibility is that this won’t happen, and the patient will die in a hospital instead. (…) this needs to be considered if dying at home is important to her.*”(FG5, RN2)

### 3.3. Balancing the Patients Medical Condition and Needs

In the sub-theme, *Acuity versus palliative needs*, discussions by both acute and SPC teams occurred on clinical decision-making, often related to the possibility to rule out an acute condition, for example, a pulmonary embolus that may be possible to treat. On the other hand, there were reflections on the uncertainty of how appropriate these actions might be, depending on the current stage and phase of the disease, which seemed difficult to assess. Often it was described, particularly by the SPC nurses, as being at a crossroad. The discussions were focusing on what could be done, rather than if, in a sub theme, *What can be done versus what should be done.* Mainly from the SPC teams, the uncertainties in clinical decision-making were reported to be related to possible ER visit/acute hospital admissions. Here, the focus was on what actions, rather than if medical actions should be taken or not. The recent improvements in cancer treatments seem to impact these decisions.

“*(…) thinking that you can always offer them (the patient/informal caregivers) to go to a hospital for an emergency assessment, and then offer them to come home as soon as possible to assess if this is an acute deterioration that may be treatable or that the patient entered another phase in the disease.*”(FG1, RN1)

Especially among the participants in the SPC teams, the uncertainty of prognosis was highlighted in, *Uncertainty of prognosis*. The expected prognosis was described in many cases to be unclear to both nurses and physicians and therefore impacted what actions that would be most appropriate. The participants highlighted situations where new cancer treatment options were available and how difficult it could be to weigh benefits against risks. The inclination to do everything possible, regardless of whether it involved invasive procedures, was explained to be stronger in today’s palliative care, compared with in the past.

### 3.4. Balancing Ethical Dilemmas

Several situations were described to cause ethical stress, especially by physicians in the acute care FGI’s. In the sub-theme *Fear of making mistakes versus Do No Harm*, it was expressed as a sense of avoiding leaving out anything that might benefit the patient, but at the same time avoiding causing harm with non-beneficial clinical decisions.

“*I am thinking of the anxiety of the situation, not to admit her to hospital and risking a dramatic death at home that the children will witness. I feel a lot of stress from this, and I think I would have chosen the “coward” way and admitted her, despite her wishes.*”(FG5, MD1)

Conflicts on what should be the next step, related to what was believed appropriate to do and the mandate to take difficult decisions were reported in the sub-theme, *Opposite views of appropriate care*. An example related to hope of positive effects from new cancer treatments and unclear individual goals of care, illustrates the discrepancies between the two organizations planned actions.

“*Maybe, maybe this new treatment will give an effect. We will see. We will do a new CT in 3 months. Then it is almost impossible for us (SPC at home) to come the next day and say that you are dying. We want to plan for this. That is tough...*”(FG2, MD1)

Ethical aspects were often connected with uncertainty, for example, *Ethical stress due to uncertainty of prognosis and goals of care.* This was described by both teams.

“*The consequences of starting too much treatment/diagnostic procedures, that may not lead anywhere, will be that she (the patient) will be in such poor condition. She might die in the hospital (…). Then you won’t be able to support and focus on the husband and the kids’ emotions, you might miss this.*” (FG3, RN3)

In summary, a lack of clarity related to responsibilities, mandate and goals-of-care is described to vary between teams, with a discrepancy between the acute cancer care and SPC at home teams regarding what was appropriate, creating ethical dilemmas

## 4. Discussion

In this study several challenges with integrating cancer care have been identified, from HCPs both in acute and palliative care, leading to uncertainty on mandate and responsibilities, risking fragmentizing the care. Organizational factors influence clinical decisions and seem to impact how shared responsibility is perceived. We mainly identified barriers related to clinical decisions, even though several suggestions of how to improve integration such as, systematic communication and improved handover procedures as well as having mutual discussions on goals of care, were reported. A qualitative study by Kaufman et al., 2020 [34], which interviewed patients and informal caregivers, suggested several similar approaches, e.g., better care coordination and support with self-care, as well as peer-support and enhanced symptom management.

A well-integrated acute and palliative cancer care is in line with international consensus [13,35]. Kaasa et al., 2018, conclude that independent of patient prognosis and treatment intention, better integration is necessary to achieve person-centered cancer care [13]. We found no apparent signs of formal integration between acute and palliative care organizations in this study, possibly a linkage working in parallel, with initiatives for integration at best on individual initiatives. On the contrary, the organizational structures seemed to rather hinder integration, resulting in SPC and acute cancer care teams working in silos, without mutual goals of care. One consequence of the uncertainty of responsibilities seemed to be clinical decision-making often steered by resources or routines (e.g., new palliative referral procedure needed after short acute admissions), rather than the individual patient’s needs.

In previous research, several studies shows that intensity of care at EOL, are increasing [4,36,37], stressing the importance of striving for better integration between SPC and acute cancer care [13]. The shift from inpatient to outpatient cancer care [38] and every transition from oncology outpatient care to SPC at home (or reverse), bring risks of missing crucial health-related information [39]. Further, health care transitions at EOL may increase not only risks of information lost in transition, but it also risks harming patients with a high symptom burden, risking a poor quality of care at EOL [39,40].

Another interesting finding in this study that was described by HCPs from both acute and palliative care, was how patients’ and their informal caregivers’ perquisites were managed. Both patient and informal caregivers were sometimes described as the single decision-maker for clinical decisions. The HCPs described the ethical dilemma as less difficult if the patient had a clear preference, making professional decisions easier. Throughout, a desire for person-centered care was apparent, but as described, in some instances made the patient the sole decision-maker. To view the patient as the “key actor”, in decision-making, is not without consequences. A study of handover processes in cancer care found that patients as “key-actors” in decision-making, often blamed themselves for the treatment outcomes putting the patients in a very vulnerable situation [41]. We can only speculate, but perhaps when HCPs experience the uncertainty of responsibilities, it increases the inclination to resign for patients and/or informal caregivers’ preferences, even if they see risks for negative consequences.

Another finding was the described balancing act between the medical condition and uncertainty of the future, focusing on the palliative versus curative phase, and HCP teams being responsible for either the cancer treatments or the symptom management. This is a clear indication of fragmentation [42], risking the poor quality of care, and lacking coordination in this highly vulnerable, often severely ill group of patients. We interpreted this because of uncertainty, both on prognosis, mandate, and lack of clear individual goals of care, with improved access to new cancer treatments, making this even more challenging.

The lack of explicit individual goals of care was highlighted as both a barrier and an enabler (in the rare occasion that clear goals were in place). Some SPC professionals stated that they were working by the “order” of the acute oncology team, without a mandate to change, question, or not knowing the goals of care. In the rare case of mutual decisions by the two teams, it was perceived as positive, making the goals clearer and more realistic. On the other hand, the acute cancer care teams described that SPC teams had the main responsibility of care decisions, as they were up to date with the patient’s general condition.

Absence or delay of clinical decisions was frequently reported, often related to treatment interruptions and/or if a patient deteriorated and seemingly led to unnecessary ER visits and unplanned hospital admissions. This is in line with findings from a Finnish study by Hirvonen et al., 2019, where no decisions were taken to stop active cancer treatment for 18% of the patients, and the decision was made at EOL in every third patient [43]. The clear discrepancy of responsibilities is worrying and could be one of the main explanations for our findings, resulting in fragmented cancer care in the region of Stockholm.

The scarcity of mutual clinical decision-making was visible in several themes. The participants from both care settings described how this resulted in patients deteriorating and ending up in acute care dying, which was described as limiting both patients and informal caregivers chances to prepare.

Across all themes, aspects of uncertainty were described and when reasoning around ethical dilemmas, the participants, were anxious about missing actions that might benefit the patient. Both nurses, and physicians expressed that the latter were increasing the likelihood of admitting the patients to acute care, even if the reason for admission would be situations that SPC was best to manage (for instance inadequate pain control or, unstable psychosocial situation).

The aspect of ethical dilemmas in our study is described in terms of being scared of making or being accused of making mistakes, and as something contributing to ethical stress in clinical decision-making at EOL, in another study on advance care planning [44].

### Strengths and Limitations

One strength of this study is that it includes health care professionals’ perspectives from both acute and palliative cancer care, as well as different organizations perspectives (non-profit, government and privately organized). From this perspective, we believe the results on the importance of integration may be transferable to other settings. In addition, using fictive patient cases may increase the transferability, since challenges in caring for patients with complex needs are not unique. On the other hand, the region of Stockholm is a large region with multiple health care providers, hence the results might not be entirely transferable to another smaller region or health care system. To increase credibility and trustworthiness, we chose an analytic method with a systematic approach, FA [30]. We found this method suitable for the aim, and foremost a way of being highly transparent in the authors pre-understanding of the topic, with early in the process organizing the data in a matrix with preliminary themes, that during the process was altered. This is particularly suitable when exploring an existing problem, rather than producing a new theory [31]. Further, FA is allowing a constant comparison between and within cases, including all data.

The authors have vast experience in oncology care and HU also in palliative care. However, none of the authors were involved in clinical decision-making related to the HCPs in the study. Although the authors’ pre-understanding could have influenced the data analyses and interpretation of the data, awareness of this risk and the continuous dialogue between the authors helped to minimize potential misinterpretations, as well as strengthening rigor. Further, as mentioned above, the FA, was used to strengthening rigor. However, the in-depth knowledge was crucial during the interviews, as it helped to understand the complexity, as well as increase the willingness of the participants to share this.

We cannot be sure if and how conducting the FGI’s face-to-face would have impacted the results. In a previous study, comparing face-to-face with interviews via a digital platform, the results indicate similar findings between the two types of interviews [45]. As this study was performed during the COVID-19 pandemic, online FDIs were the only possible option as no external visitors including researchers were allowed in any health care setting during this time-period.

A common recommendation in qualitative research is to invite six to eight participants in focus groups [46]. In this study, the number of participants varied between two (one FGI) and five (one FGI). Slightly smaller groups have also been recommended, as it may facilitate discussing sensitive topics and also increase the involvement of all participants [47]. Most of the participants were nurses and women, which may have influenced the results. However, this also reflects the reality, as nurses are the largest professional group in health care [48] and health care is heavily female-dominated.

We have not yet provided the findings to the participants; however, we have presented the preliminary findings at several forums, and asked for feedback from HCP working in the same setting as the participants. In general, our preliminary findings have been well-recognized among both nurses and physicians, and we perceive a great interest among the HCP to further discuss and act on those findings

## 5. Conclusions

There are clear uncertainties related to both responsibilities, organization, and mandate with and between cancer care teams caring for patients with complex care trajectories. These uncertainties may result in negative consequences for the patients, the informal caregivers, HCPs, and the health care organizations. We suggest policymakers and other stakeholders focus on the integration of care on all levels and to build a system steered primarily by the quality of care. Implementation of person-centered integrated care, with individual care plans including goals, and clear routines for systematic communication between cancer care teams, patients, and informal caregivers, regardless of organization should be prioritized, to avoid fragmentation of care. More research is needed in this field, from other regions and countries, to explore the impact of different health care systems, but also to include the patient and informal caregiver’s perspectives.

## Figures and Tables

**Figure 1 ijerph-19-16134-f001:**
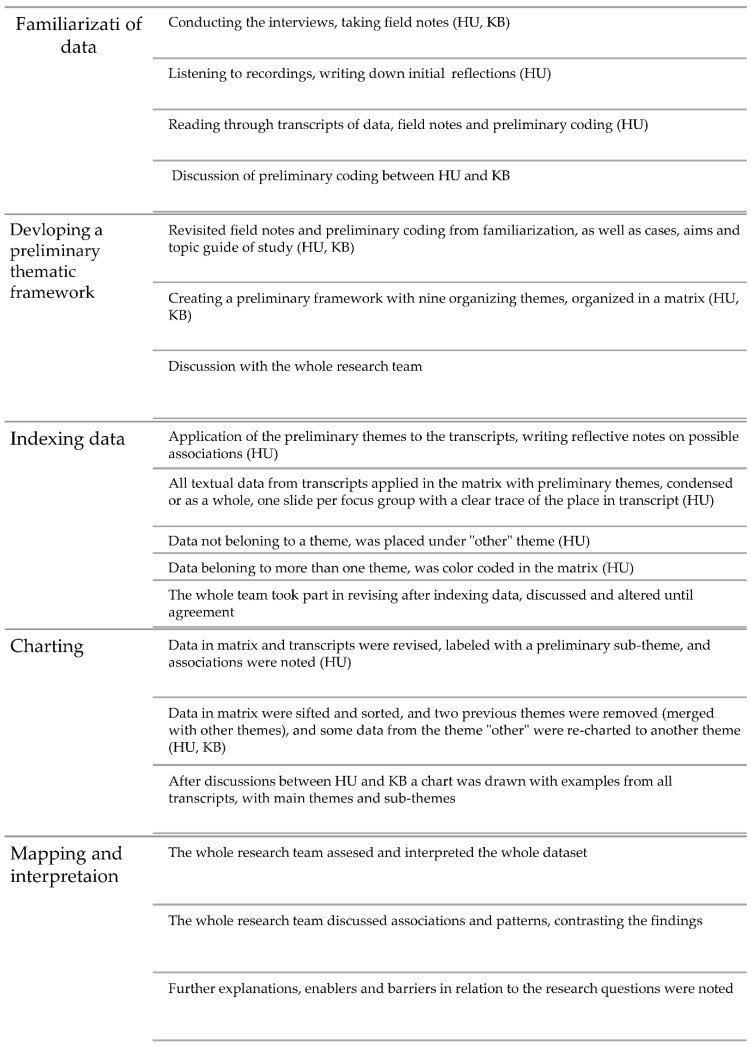
Description of the analytical process according to the FA approach [30,31].

**Figure 2 ijerph-19-16134-f002:**
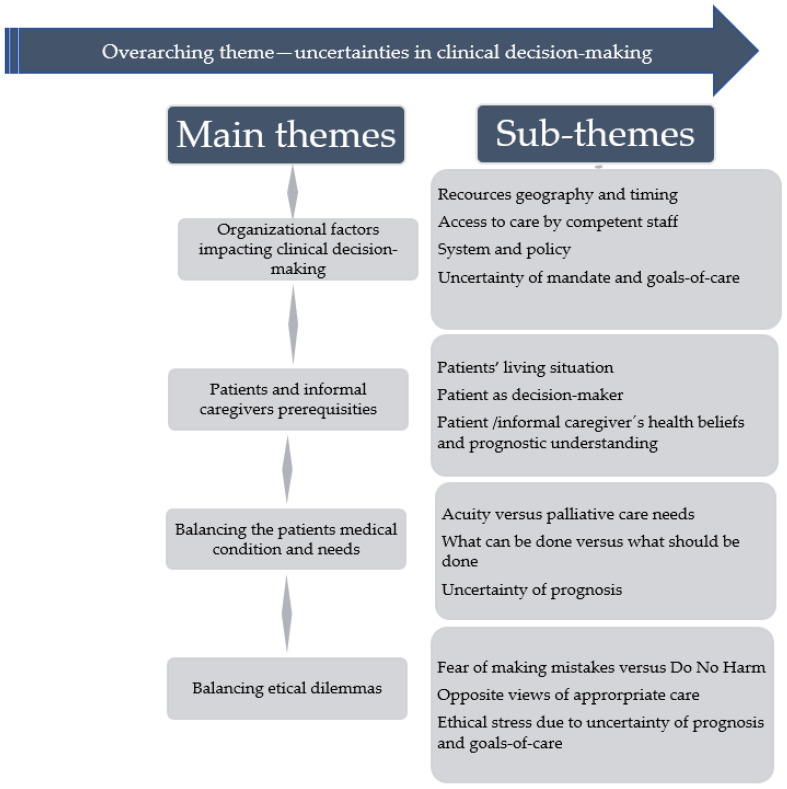
The overarching theme, main themes, and sub-themes.

**Table 1 ijerph-19-16134-t001:** Characteristics of participants in the focus groups.

Characteristics of the Participants	N = 22
Gender	
Men	4
Women	18
Age groups	
20–40 years	7
41–60 years	15
Profession	
Nurse	14
Physician	8
Years in profession	
0–5 years	4
6–15 years	11
>15 years	7
Workplace	
SPC at home	16
Acute cancer care	6
Specialization	
Oncology	8
Geriatrics and/or palliative care	1
Not specified	2
Not specialized	11
Years at current workplace	
0–5 years	6
6–15 years	13
>15 years	3

## Data Availability

The authors believe all relevant data are among the manuscript. Data cannot be shared publicly because of rules and regulations in Sweden on research data, data is stored without a personal identification no to protect autonomy, this according to the ethics approval with no DNR 2020-00910. The authors are not allowed to publicly share any of the data. Any requests for data must always go through the Swedish ethics board. They may be contacted here: registrator@etikprovning.se, +46-(0)10-475-08-00.

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
