# Peer review of "Exploring Health Care Professionals’ Perceptions Regarding Shared Clinical Decision-Making in Both Acute and Palliative Cancer Care"

_ijerph, 2022, doi:10.3390/ijerph192316134_

Round 1

Reviewer 1 Report

The submitted article for review based on qualitative explorative design, is a very interesting analysis. Therefore, it is worth mentioning in the summary: the number of interviews conducted; professions (nurses and physicians) instead of „ health care professionals”; place (country); when they were carried out. Also the Discussion section would need a small restructuring. The reader needs to be able to distinguish between statements that the authors have made based on the results vs. assumptions (new suggestions that the authors derive on the basis of the results, that could be tested in future research). The authors described the limitations of the study well.

I suggest that this article deserves publication.

Author Response

Reviewer 1

The submitted article for review based on qualitative explorative design, is a very interesting analysis. Therefore, it is worth mentioning in the summary: the number of interviews conducted; professions (nurses and physicians) instead of „ health care professionals”; place (country); when they were carried out. 

Thank you for pointing this out, we have addes some more info in summary (abstract). As well as clarified what we mean with health care professionals in line 86.

Also the Discussion section would need a small restructuring. The reader needs to be able to distinguish between statements that the authors have made based on the results vs. assumptions (new suggestions that the authors derive on the basis of the results, that could be tested in future research).

Thank you for this, throughout the discussion, we tried to be clearer with this and made small adjustments.

The authors described the limitations of the study well.

I suggest that this article deserves publication.

Thank you for encouragment and your valuable comments.

Please note that all changes are with tracked changes/and-or red, according to the journals request for revised manuscripts.

Since reviewers sugggested alterations of figure 1 & 2, the old versions are still visible, as well as the new figures, however the old figure 1 & 2 are marked to delete with tracked changes.

Reviewer 2 Report

Thanks to the authors for the possibility to review this article. I think the topic is very important. I agree with the authors that more research is needed in this field. The number of participants in the study is low, but despite this the study has important informative value. I recommend small corrections to the formal form of images 1 and 2. Especially the image 2 is very large and has different fonts. It would be better if the figures had the similar form to the tables.

Author Response

Thanks to the authors for the possibility to review this article. I think the topic is very important. I agree with the authors that more research is needed in this field. The number of participants in the study is low, but despite this the study has important informative value.

Thank you for this, and your suggestions, see below.

I recommend small corrections to the formal form of images 1 and 2. Especially the image 2 is very large and has different fonts. It would be better if the figures had the similar form to the tables.

Thank you for pointing this out! We have altered Figure 1-2 accordingly and agree this is an improvement.

Please note that all changes are with tracked changes/and-or red, according to the journals request for revised manuscripts.

Since reviewers sugggested alterations of figure 1 & 2, the old versions are still visible, as well as the new figures, however the old figure 1 & 2 are marked to delete with tracked changes.

Reviewer 3 Report

this is a very interesting well crafted and well presented analysis on uncertainties regarding clinical decision making in acute /palliative cancer care

I have only one recommendation which is to change the current title with 

Exploring healthcare professionals uncertainties regarding .... in this way it better reflects the content. And if the analysis explored some other perceptions make sure to draft the results in the future, 

Author Response

this is a very interesting well crafted and well presented analysis on uncertainties regarding clinical decision making in acute /palliative cancer care

Thank you! Pls see below!

I have only one recommendation which is to change the current title with 

Exploring healthcare professionals uncertainties regarding .... in this way it better reflects the content. And if the analysis explored some other perceptions make sure to draft the results in the future, 

Thank you for pointing this out, we have altered accordingly and agree that this is better reflection of the study´s content.     

Please note that all changes are with tracked changes/and-or red, according to the journals request for revised manuscripts.

Since reviewers sugggested alterations of figure 1 & 2, the old versions are still visible, as well as the new figures, however the old figure 1 & 2 are marked to delete with tracked changes.

Reviewer 4 Report

Thank you for the opportunity to review this paper that contributes knowledge about healthcare professionals’ perceptions on decision-making in the field of cancer care.

Despite the manuscript being well written, there are several methodological issues that need to be addressed to strengthen rigour and make the work suitable for publication. Suggestions are given below.

Major

-        Lines 96-101. Authors stated that the fictive patient cases were tested for face validity by healthcare professionals. How many healthcare professionals participated? What was their scope in practice? How were they approached? Were any changes made after feedback from healthcare professionals?

-        Lines 101-104. Were the prompting questions related to the case pilot-tested?

-        Line 113. The authors reported that 90 minutes were allocated for focus group interviews. Did they all last 90 minutes? Or the duration varied in some cases? Please specify the median duration with range.  

-        Were the interview transcripts checked for accuracy? If so, how?

-        Line 116: When were the field notes taken? During and / or after the focus group interviews? Moreover, it is unclear how the field notes were used in the analytical process. They are also not mentioned in the results section. This requires more attention.

-        Line 117: What do the authors mean by ‘experienced medical administrator’?  Did he/she join the focus group interviews or just transcribe the recordings? Also, were the transcripts returned to participants to check for errors or additional comments?

-        Who coded the data and how? Were two independent researchers involved? How did the coding process work?

-        It is not clear how the preliminary framework with themes organized in a matrix was created. More information is needed.  It may be useful to have the original matrix and the final matrix in the appendix. This would add transparency to the analytical process.

-        Was a software used to aid the analysis? If so, which one?

-        Information on data saturation lack, please add.

-        How the overarching theme emerged through the analytical process remains blurred and needs clarification.

-        Did the participants provide feedback on the findings?

-        How did the authors pursue rigour and trustworthiness? This section is missed in the methods and should be added.  

Minor

-        Figure 1. Several typos. Transripts and transripts instead of transcripts. Familirazation instead of familiarization. Associastions instead of associations. Frameork instead of framework. Themese instead of themes. Esearch instead of research.

-        Figure 2. Typo, perequisities instead of prerequisites.

-        Line 269. Grammatical error.

-        Consistency in acronyms associated with quotations is required. For example, in some cases it is reported RN while in others nurse. Moreover, it would be useful to know when quotations belong to SPC teams members or to acute cancer teams members.

-        Line 278. What does P4 mean?

-        Line 384. Typo, perquisites.

-        Line 404. Two full stops instead of one.

-        Line 443. A verb is likely missed.

-        The bibliography needs extensive revision. Some references miss the doi number. Furthermore, pages do not appear in the first reference. The paper cited in reference n°25 was published in 2021, not 2020 and pages are lacking.  

-        Grammar and punctuation need revision throughout the entire manuscript.

Author Response

       Lines 96-101. Authors stated that the fictive patient cases were tested for face validity by healthcare professionals. How many healthcare professionals participated? What was their scope in practice? How were they approached? Were any changes made after feedback from healthcare professionals? Lines 101-104. Were the prompting questions related to the case pilot-tested?

Thank you for pointing this out, we have in line 100-106 clarified this. We are clarifying that also the prompting questions were included here, as well as in a test interview performed.

-      Line 113. The authors reported that 90 minutes were allocated for focus group interviews. Did they all last 90 minutes? Or the duration varied in some cases? Please specify the median duration with range.  

Thank you for pointing this out, and we have clarified in line 140-141.

-     

       Were the interview transcripts checked for accuracy? If so, how?

       The first author, listened to all recordings and simultaneously reviewed the transcripts, and field notes.

-        

       Line 116: When were the field notes taken? During and / or after the focus group interviews? Moreover, it is unclear how the field notes were used in the analytical process. They are also not mentioned in the results section. This requires more attention.

Line 117: What do the authors mean by ‘experienced medical administrator’?  Did he/she join the focus group interviews or just transcribe the recordings? Also, were the transcripts returned to participants to check for errors or additional comments?

       Thank you for noticing this. With experienced medical administrator, we mean a medical secretary, working in a hospital, familiar with medical terminology, and especially with transcribing. This person was not participating during interviews, and we did not return the transcripts to the participants to check, mainly because this wasn’t considered as feasible, out of respect of their time.

       Who coded the data and how? Were two independent researchers involved? How did the coding process work?

       This was not clear in the methods section, thank you for pointing this out to us. We have clarified in Figure 1, by noting the initials of the involved authors for each step, to make the process more transparent.

       It is not clear how the preliminary framework with themes organized in a matrix was created. More information is needed.  It may be useful to have the original matrix and the final matrix in the appendix. This would add transparency to the analytical process.

We have added an additional appendix B (directly after Appendix A), line xx with a figure who displays the process, how the preliminary framework was altered during the process, we hope this makes the process clearer.

        Was a software used to aid the analysis? If so, which one?

        No software was used. Microsoft excel was used to aid the analysis.

        Information on data saturation lack, please add.

     Thank you for pointing this out, it is important. We have added clarification on this in line     135-136.

       How the overarching theme emerged through the analytical process remains blurred and needs clarification.

       We have tried to clarify this in line 160-161, thank you for pointing this out, we hope this is clearer now.

       Did the participants provide feedback on the findings?

       No, we have not yet provided the findings to the participants, however we have presented the preliminary findings at several forums, asked for feedback from HCP working in the same setting as the participants.

       How did the authors pursue rigour and trustworthiness? This section is missed in the methods and should be added. 

       Thank you for highlighting this important matter. We choose to elaborate on how we did this in the discussion (under the heading “Strength and limitations”. Here we had previously discussed both rigor and trustworthiness, however not clearly enough. Please see lines 452-453 as well as lines 464-465 for clarification.

Minor

       Figure 1. Several typos. Transripts and transripts instead of transcripts. Familirazation instead of familiarization. Associastions instead of associations. Frameork instead of framework. Themese instead of themes. Esearch instead of research.

       Thank you for this and we have corrected the typos in figure 1.

       We suspect that we had a problem with spelling/grammar check perhaps using the template, however we have now corrected typos and checked the whole manuscript again.

      Figure 2. Typo, perequisities instead of prerequisites.

      Thank you for this, this is now corrected.

       Line 269. Grammatical error. Corrected

      Consistency in acronyms associated with quotations is required. For example, in some cases it is reported RN while in others nurse. Moreover, it would be useful to know when quotations belong to SPC teams members or to acute cancer teams members.

       We have revised this throughout the quotations, thank you for noticing. In addition, we tried to clarify within the text in the manuscript, but due to keeping confidentiality and autonomy,  we don’t want to specify directly in the quotes from which team the person we are quoting comes.  

       Line 278. What does P4 mean?

       This means from what page, but is removed now to be consistent, since not relevant in the manuscript.

       Line 384. Typo, perquisites. Corrected

       Line 404. Two full stops instead of one. Corrected

       Line 443. A verb is likely missed. Corrected

       The bibliography needs extensive revision. Some references miss the doi number.

 Furthermore, pages do not appear in the first reference.

We used the recommended format according to author guidelines (MDPI ACS) and revised the whole reference list.

The paper cited in reference n°25 was published in 2021, not 2020 and pages are lacking.

This paper was published online in 2020, when using citation manager (EndNote) this is the correct year.    

      Grammar and punctuation need revision throughout the entire manuscript.

      Thank you for this! We have gone through grammar and punctuation.

Please note that all changes are with tracked changes/and-or red, according to the journals request for revised manuscripts.

Since reviewers sugggested alterations of figure 1 & 2, the old versions are still visible, as well as the new figures, however the old figure 1 & 2 are marked to delete with tracked changes.

Round 2

Reviewer 4 Report

Thank you for going across most of the previous comments. Anyway, some major concerns have not been adequately addressed. Particularly:

-     - When were the field notes taken? During and/or after the focus group interviews? Moreover, it is unclear how the field notes were used in the analytical process. They are also not mentioned in the results section. This requires more attention.

-          All high-quality qualitative papers describe in the methods section how rigour and trustworthiness have been pursued. This is mandatory to guarantee transparency and increase the readers’ confidence in the results of the study. The authors can then further comment on these issues in the section of strengths and limitations.

-      - It continues not to be clear to me how the overarching theme emerged through the analytical process. This should be clarified in the methods section.  

Minor points

-      - Some typos still appear. For example, in Appendix B “framwork”, “perequisities”. Please, check.

-    - Some previous comments have been addressed only in the authors’ response but without resulting in changes in the main manuscript (eg., What do the authors mean by ‘experienced medical administrator’? Did he/she join the focus group interviews or just transcribe the recordings? Also, were the transcripts returned to participants to check for errors or additional comments?; Was a software used to aid the analysis? If so, which one?; Did the participants provide feedback on the findings?). It would be worth adjusting the main text according to the responses already provided.

-      - Appendix B should be referenced in the main text.

Author Response

Thank you please se attached file with response. 
